# Evaluation of Hepatic Biochemical Parameters during Antiviral Treatment in COVID-19 Patients

**DOI:** 10.3390/biology11010013

**Published:** 2021-12-23

**Authors:** Felicia Marc, Corina Moldovan, Anica Hoza, Patricia Restea, Liliana Sachelarie, Laura Ecaterina Romila, Corina Suteu, Dorina Maria Farcas

**Affiliations:** 1Clinical Departament, Faculty of Medicine, Oradea University, 410610 Oradea, Romania; feliciamarc.dr@gmail.com (F.M.); cmold2003@yahoo.com (C.M.); anica_hoza@yahoo.com (A.H.); resteapatri@yahoo.com (P.R.); suteucorina@gmail.com (C.S.); dmfarcas@yahoo.com (D.M.F.); 2Preclinical Department, Apollonia University, 700535 Iasi, Romania

**Keywords:** coronavirus, antivirals, hepatic parameters, COVID-19

## Abstract

**Simple Summary:**

In order to treat COVID-19 disease, various drugs have been used as repurposed drugs, because no drug directly targets against the SARS-CoV-2 virus. The aim of this study was to evaluate the relationship between the drugs used for COVID-19 treatment and liver disturbances, in order to identify any change in liver enzymes during therapy. Patients admitted in an internal medicine department were treated with a complex therapeutic scheme, including antivirals. Beside the follow up for the evolution of the disease, we also monitored the potential occurrence of side effects, especially liver damage. Our results showed that none of the three antivirals that we used produced severe or significant liver disturbances. Our conclusion may be useful in guiding clinical practice, adding more information for the medical community.

**Abstract:**

(1) Background: The antiviral treatment for COVID-19 disease started to be largely used in 2020 and has been found to be efficient, although it is not specific for SARS-CoV-2 virus. There were some concerns that it may produce liver damage or other side effects. (2) Methods: The aim of this study was to observe if antiviral therapy is affecting liver parameters or producing other side-effects in patients hospitalized for COVID-19 disease. The study included a group of patients hospitalized in the internal medicine department of Oradea Municipal Clinical Hospital, Romania, between August 2020–June 2021, diagnosed with SARS-CoV-2 viral infection by RT-PCR method or rapid antigen test. During hospitalization, patients were treated with a Lopinavir/Ritonavir (Kaletra) combination, or with Favipiravir or Remdesivir. In addition to monitoring the evolution of the disease (clinical and biochemical), also hepatic parameters were analyzed at admission, during hospitalization, and at discharge. (3) Results: In the group of studied patients, the mean value of aspartat aminotrensferase did not increase above normal at discharge, alanin aminotransferase increased, but below twice the normal values, and cholestasis registered a statistically insignificant slight increase. (4) Conclusions: In our study, we found that all three antivirals were generally well tolerated and their use did not alter liver function in a significant manner.

## 1. Introduction

Infection with the new coronavirus is currently the most important topic worldwide [1,2,3,4]. The emergence of COVID-19 disease implies collaboration and an interdisciplinary effort for its understanding and characterization [3].

In practice, the study of patients with COVID-19 evidenced a different evolution from individual to individual, with only a series of common manifestations [5,6].

The mechanism of action for coronavirus, as well as its appropriate treatment, is still being studied [7]. Recent studies indicate that mortality with COVID-19 is also associated with increased cardiovascular risk, coagulopathy, and disseminated intravascular coagulation [8,9,10].

The treatment plan for viral or mixed (viral and bacterial) pneumonia includes antiviral, anticoagulant, anti-inflammatory (Dexamethasone), and antibiotic drugs [11,12,13]. In the clinical forms in which cytokine storm occurs, the treatment plan is supplemented with immunomodulators (Tocilizumab or Anakinra) [14,15,16].

The 2019 coronavirus disease pandemic (COVID-19), due to the new severe acute respiratory syndrome (SARS-CoV-2), has led worldwide to a sharp increase in hospitalization of patients with pneumonia and multiorgan diseases [16,17,18,19].

The antiviral treatment started to be largely used in 2020, although it is not specific for SARS-CoV-2 virus. From a chronological point of view, at the beginning a combination of two antiretrovirals was used Lopinavir/Ritonavir, then Remdesivir and Favipiravir.

However, there were some concerns about potential liver damage or other side effects.

Especially the combination of Lopinavir/Ritonavir was strongly criticized for not being effective against COVID-19 disease and having side effects, one of them being the alteration of hepatic function. In our department of internal medicine we had many patients who refused this therapy and signed in their medical file. Also there were some reserves for Favipiravir and Remdesivir, as they were newly introduced drugs, with not too much previous experience in coronavirus disease.

The aim of the present study was to evaluate the hepatic function, hepatocytolysis, and cholestasis syndrome, and full blood count with neutrophils and platelets counts during treatment with antiviral medication, Ritonavir/Lopinavir (Kaletra), Favipiravir, and Remdesivir, from the beginning of hospitalization, during the treatment, and at discharge.

## 2. Materials and Methods

The research was focused on 272 patients diagnosed with SARS-CoV-2 viral infection, treated in the Internal Medicine Clinical Department of the “Gavril Curteanu” Municipal Clinical Hospital of Oradea, Romania, between 1 August 2020–7 June 2021 (date of discharge of the last COVID-19 patient from the Internal medicine department) diagnosed with SARS-CoV-2 viral infection by the RT-PCR method and subsequently, when it became available, by the rapid antigen test.

Among the study group, 36 patients were treated with Lopinavir/Ritonavir (Kaletra) (L/R), 85 patients with Remdesivir, and 151 patients with Favipiravir. L/R treatment was subsequently withdrawn from the protocol of the Ministry of Health; there were also patients who refused to receive this treatment combination. Patients with significant pre-existing liver damage (chronic hepatitis with hepatocytolysis or severe cholestasis, liver cirrhosis, and liver metastases) were excluded from the study. The results of the computer tomography (CT) scan of the abdomen were also evaluated to rule out causes of liver damage.

In order to establish the disease’ severity, patients were evaluated from a clinical point of view and also with biochemical analysis and thoracic, abdominal, and cranial CT (as needed). From the clinical point of view, mild forms were with no oxygen requirements, while severe forms required high flow oxygen support.

From the paraclinical point of view, to determine the severity of the disease, whether its mild, moderate, or severe, and also if it is viral or mixed COVID-19 pneumonia (viral and bacterial), the most important criteria was the degree of lung involvement described at computer tomography and the value of biochemical parameters characteristic for this coronavirus (C reactive protein, ESR = erythrocite sedimentation rate, lactate dehydrogenase, creatinkinase, limphocytes, plateletes, and D-dimers). Also, damage to other organs and systems caused by coronavirus and the presence of co-morbidities were used in order to classify the severity of the disease.

Subsequently, drug treatment was initiated, in accordance with the protocols of the Ministry of Health in force at that time and was adapted to every clinical forms of the disease.

The treatment was based on administration of antiviral (Lopinavir/Ritonavir—Kaletra, Remdesivir—Veklury, Favipiravir—Fluguard), anticoagulant, steroidal anti-inflammatory, antibiotic, as well as immunomodulators drugs, in the case of clinical and biochemical aggravation in the 8–12 days of the disease (cytokine storm), with favorable clinical evolution in most cases.

In the Ministry of Health protocol, the potential side effects of antiviral drugs are the following—for the combination Lopinavir/Ritonavir: hepatocytolysis syndrome/cholestasis, leukopenia, nausea, and vomiting; for Remdesivir: hepatocytolysis syndrome/cholestasis, thrombocytopenia, thrombocytopenia; and for Favipiravir: hyperuricemia, diarrhea, hepatocytolysis/cholestasis syndrome, and thrombocytopenia.

The monitored biochemical parameters were the following:-for Favipiravir, hepatocytolysis and cholestasis syndrome (GOT, GPT, GGT, ALP, and total as well as uric acid value)-for Remdesivir, hepatocytolysis and cholestasis syndrome (GOT, GPT, GGT, ALP, and total as well as the value of neutrophils and platelets)-for Lopinavir/Ritonavir, hepatocytolysis syndrome, cholestasis, as well as the value of neutrophils and platelets.

The value of these parameters was determined initially (at hospitalization), during treatment, and on the day before discharge.

The results of the CT scan of the abdomen were also evaluated in order to rule out causes of pre-existing liver damage (chronic viral hepatitis, liver cirrhosis, liver metastases, etc.). All subjects gave their informed consent for inclusion before they participated in the study, conducted in accordance with the Declaration of Helsinki.

Data processing was performed using the SPSS 20 program. Frequency ranges, average parameter values, and standard deviations were calculated. Tests of statistical significance by the χ2 method were used, and ANOVA (Brown-Forsythe) was used to compare the means. The level of statistical significance was 0.05.

## 3. Results

### 3.1. Characteristics of the Population (Sex, Age, and Environment)

In our study group, over 50% were women (51.10%), the ratio of women/men was 1: 1, the age was between 26–88 years, the average age was 60.18 years, and the patients came mainly from the urban environment (55.88%); (Table 1) the urban/rural ratio was 1.3:1.

In the Lopinavir/Ritonavir (Kaletra) (L/R) batch, men predominated (58.33%), the percent being insignificantly higher than in the Remdesivir and Favipiravir groups (48.24%, *p* = 0.312, respectively 47.02%, *p* = 0.224).

### 3.2. The Severity of the Disease

The severe form of the disease was found in 82.35% of patients from the Remdesivir batch, significantly higher than in the L/R and Favipiravir groups (41.67% and 54.97%, respectively, *p* < 0.001), (Table 2, Figure 1).

### 3.3. Evolution of Hepatic Function Parameters

The normal laboratory GOT values are between 0 and 50 U/L.

Slightly increased values at hospitalization were interpreted as occurring in the context of the disease, the SARS-CoV-2 virus also causing liver damage.

In the L/R and Favipiravir groups, the GOT values had a decreasing trend, and in the case of Remdesivir, they had a convex evolution (Table 3).

Thus, compared to the values at hospitalization, in the L/R and Favipiravir groups, the GOT values decreased insignificantly after administration (from 56.58 U/L to 45.65 U/L, *p* = 0.142 and from 51.72 U/L to 45.48 U/L, *p* = 0.066, respectively), further decreasing, so that, at discharge, the decrease in the case of Kaletra batch was insignificant (from 56.58 U/L to 44.26 U/L, *p* = 0.112), whereas the decrease in the Favipiravir batch was significant (from 51.72 U/L to 40.35 U/L, *p* < 0.001).

In the Remdesivir batch, GOT values increased insignificantly after administration (from 51.11 U/L to 52.12 U/L, *p* = 0.834) and decreased insignificantly at discharge from 51.11 U/L to 45.12 U/L, *p* = 0.231).

The comparison of the three groups, at hospitalization, during administration, and at discharge shows no significant differences in GOT values (*p* > 0.05), while the use of each of the three drugs did not lead to increased GOT (Figure 2).

The normal laboratory GPT values are between 0 and 50 U/L.

In all three groups, GPT values followed an increasing curve at hospitalization, after administration, and at discharge. In the L/R batch, the GPT values at hospitalization were 47.67 U/L, after administration of 52.85 U/L (*p* = 0.477), and at discharge 60.94 U/L (*p* = 0.114), (Table 4).

Comparatively to the baseline in the Remdesivir and Favipiravir batch, GPT increased significantly after administration (from 45.47 U/L to 57.47 U/L, *p* = 0.042, and from 48.9 U/L to 59.39 U/L, *p* = 0.025, respectively), at discharge the GPT values being significantly higher (from 45.47 U/L to 66.53 U/L, *p* = 0.003 and from 48.99 U/L to 59.67 U/L, *p* = 0.022, respectively).

Comparing the three study groups, at hospitalization, after administration, and at discharge, no significant differences were observed in terms of GPT values (*p* > 0.05). In each of the administered drugs, GPT at discharge was higher than normal, but below 2x the normal value. At discharge, hepatoprotective treatment was prescribed to these patients (Figure 3).

The normal laboratory GGT values are between 0 and 50 U/L. GGT values were recorded only in Remdesivir and Favipiravir groups, at hospitalization and at discharge. In both groups, the GGT increased insignificantly (from 59.14 U/L to 62.59 U/L in the Remdesivir group, *p* = 0.633 and in the case of Favipiravir from 81.45 U/L to 97.01 U/L, *p* = 0.437, respectively), (Table 5).

At hospitalization, GGT values were slightly higher in the Favipiravir batch (81.45 U/L versus 59.14 U/L, *p* = 0.055). Both Remdesivir and Favipiravir treatments cause a slight increase in GGT (cholestasis), (Figure 4).

The normal laboratory neutrophils values are between 1800 and 7500/mm^3^. In all three batches, they followed an increasing curve at hospitalization, after administration, and at discharge.

In the L/R batch, the values of neutrophils at hospitalization were 4.43 × 10^3^/dL; after administration, they increased significantly to 8.22 × 10^3^/dL (*p* <0.001), and at discharge the values were 8.62 × 10^3^/dL (*p* < 0.001) (Table 6).

In the Remdesivir batch, the values of neutrophils at hospitalization were 6.74 × 10^3^/dL, after administration they increased significantly to 8.98 × 10^3^/dL (*p* = 0.010), and at discharge the values were 12.82 × 10^3^/dL (*p* < 0.001).

In the Favipiravir group, the values of neutrophils at hospitalization were 5.55 × 10^3^/dL, after administration they increased significantly to 7.90 × 10^3^/dL (*p* < 0.001), and at discharge the values were 8.10 × 10^3^/dL < 0.001).

Comparison of the three groups shows that, at hospitalization, the values of neutrophils were 4.43 × 10^3^/dL, significantly lower than in the Remdesivir and Favipiravir batch (6.74 × 10^3^/dL, *p* = 0.002, respectively 5.55 × 10^3^/dL, *p* = 0.034).

After administration, there were no significant differences among the three groups, while at discharge, the highest neutrophil values were recorded in the Remdesivir batch (12.82 × 10^3^/dL), significantly higher than in the Lopinavir/Ritonavir and Favipiravir batch (8.62 × 10^3^/dL, *p* = 0.007, respectively 8.10 × 10^3^/dL, *p* = 0.001). The patients analyzed in this study did not develop leukopenia using these three antiviral drugs (Figure 5).

The normal laboratory platelets values are between 150,000–400,000/mm^3^.

In the L/R batch, platelet values had a concave evolution, increasing after administration, and decreasing at discharge, while in the Remdesivir and Favipiravir batches, the trend was increasing. These variations did not exceed normal values.

Compared to the initial evaluation, in all three batches, the values increased significantly after administration (from 208.20 × 10^3^ μL to 316.59 × 10^3^ μL, *p* < 0.001; from 237.69 × 10^3^ μL to 294.12 × 10^3^ μL, *p* = 0.001; and from 231.83 × 10^3^ μL to 283.74 × 10^3^ μL, *p* < 0.001) and remained significantly higher at discharge from 208.20 × 10^3^ μL to 286.59 × 10^3^ μL, *p* = 0.004; from 237.69 × 10^3^ μL to 296.53 × 10^3^ μL, *p* < 0.001; and from 231.83 × 10^3^ μL to 297.99 × 10^3^ μL, *p* < 0.001), (Table 7).

At hospitalization, after administration and at discharge, no significant differences were observed in all three groups in terms of platelet values (*p* > 0.05). The use of the three antiviral drugs did not cause thrombocytopenia (Figure 6).

CT images of the liver revealed changes only in the Remdesivir and Favipiravir groups (10.58% vs 5.96%, *p* = 0.199) (Table 8, Figure 7).

## 4. Discussion

Liver plays a major role in many diseases and proves to be important also in COVID-19. There are several theories proposed for liver damage in COVID-19, such as direct effect of the virus on hepatocytes or biliary epithelium via Angiotensin-converting enzyme (ACE2) receptors expression, liver injury related to increased immune response (Cytokine storm) and immune-mediated damage, drug toxicity (Acetaminophen, antivirals or Hydroxychloroquine), and liver failure occurring in patients having multiorgan dysfunction [20,21,22,23,24].

In our study we excluded patients with acute liver injury and elevated liver enzymes at admission, these findings being associated with increased severity and poor outcome. The drugs used in treating COVID-19 disease were initially intended for other diseases (viruses). Various drugs are being used as repurposed drugs, as there is no specific drug or effective treatment strategy against COVID-19. Multiple challenges associated with repurposed drugs have been identified, including dose adjustments, route of administration, and acute/chronic toxicity. The relationship between the drugs used for COVID-19 treatment and liver disturbance remains controversial. It is essential to evaluate the potential liver damage caused by various drugs in order to help guide clinical practice [25,26,27,28,29].

In our study, antiviral treatment was offered to an approximately equal number of women and men (139 women and 133 men), with a mean age of 59.95 years.

The low number of patients has several explanations; for example, there was a strong opinion in the population that the combination Lopinavir/Ritonavir is not effective against the coronavirus SARS-CoV-2, and some patients refused the administration of this drug and signed in their consent. Also, there were some reserves against Favipiravir and Remdesivir, because they were newly introduced drugs; another serious problem was that patients arrived at the hospital after 10 or more days of treatment at home, and antiviral therapy was not useful anymore. Patients with high levels of aminotransferases were excluded from the study.

Chronologically, Lopinavir/Ritonavir (Kaletra), Remdesivir, and Favipiravir treatments were administrated to patients with COVID-19 pneumonia. The treatment was performed according to the recommendations of the Ministry of Health protocols in force at that time. The Lopinavir–Ritonavir treatment was withdrawn from the protocol, but the other two drugs remained as a recommendation. Remdesivir was used for the treatment of severe forms of the disease, and Favipiravir for mild and moderate forms.

Due to warnings that antiviral therapy may lead to impaired liver function, increased attention has been provided to hepatic parameters during the treatment including the three antiviral drugs, therefore, parameters indicating hepatocytolysis and cholestasis have been monitored more frequently.

In a randomized controlled study on Lopinavir/Ritonavir-treated adult patients hospitalized with mild/moderate disease, only one patient developed elevation over 2.5-fold above the normal limit [30].

In our study, GOT (aspartat aminotransferase) did not increase and GPT (alanin aminotransferase) increased from a median value of 47.67 U/L to 60.94 U/L. Also, this combination does not influence the values of neutrophils and platelets.

The association of Remdesivir with liver injury remains uncertain and is essential to evaluate the safety of this drug. One randomized multicenter trial with 237 patients [31] found similar aminotransferases levels between Remdesivir and placebo group. In our study, GOT did not increase, but the value of GPT increased from a median value of 45.4 U/L to 66.5 U/L. We also observed a slight elevation of cholestasis—gammaglutamil transpeptidase increased from 59.1 to 62.5 U/L and had no negative effect on the number of neutrophils and plateletes.

In a recent article [32], it was stated that it is important to evaluate the potential liver damage caused by various drugs in order to help guide clinical practice. In this review, these treatments were associated with minimal liver function abnormalities, but it is very important to pay attention to multimedication. In fact, we found references about Lopinavir/Ritonavir and Remdesivir, but few references about Favipiravir. Also, we made a comparison between the three antivirals and the effect of each one on liver parameters.

In our study, Favipiravir did not alter the value of GOT; GPT registered a change from 48.9 U/L to 59.6 U/L and also cholestasis enzyme increased—from 81.4 to 97.0 U/L. The levels of neutrophils and platelets were not affected.

## 5. Conclusions

From the viewpoint of the hepatic function damage (hepatocytolysis syndrome), the treatment with the three antiviral drugs did not lead to an increase in GOT values. Instead, the value of GPT increased after the use of each of the three drugs, but only slightly; no doubling of the values was recorded with any of the drugs. We observed that Remdesivir has the highest influence on the value of GPT, but not in a dangerous manner. In fact, no patient was withdrawn from the therapy.

Regarding the cholestasis syndrome, the GGT value increased slightly, but statistically insignificant, under the influence of treatments with Favipiravir and Remdesivir. Favipiravir caused a more pronounced cholestasis syndrome than Remdesivir.

The parameters related to the full blood count—the neutrophils and platelets, which could have been modified by the antiviral treatment—were not negatively influenced by the use of the three drugs, so no neutropenia or thrombocytopenia was observed.

The use of the three antiviral drugs did not cause major liver damage and, clinically, in most cases, the COVID-19 viral pneumonias had a favorable evolution.

These findings may be useful for the medical community, offering trust and adding more information about the fact that these antiviral drugs, although not specific, are not as dangerous as they seemed to be at the beginning of the pandemic.

## Figures and Tables

**Figure 1 biology-11-00013-f001:**
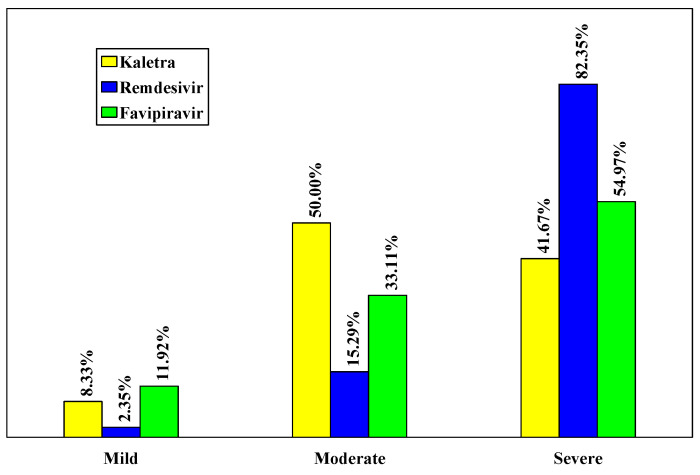
Distribution of cases according to the severity of the disease.

**Figure 2 biology-11-00013-f002:**
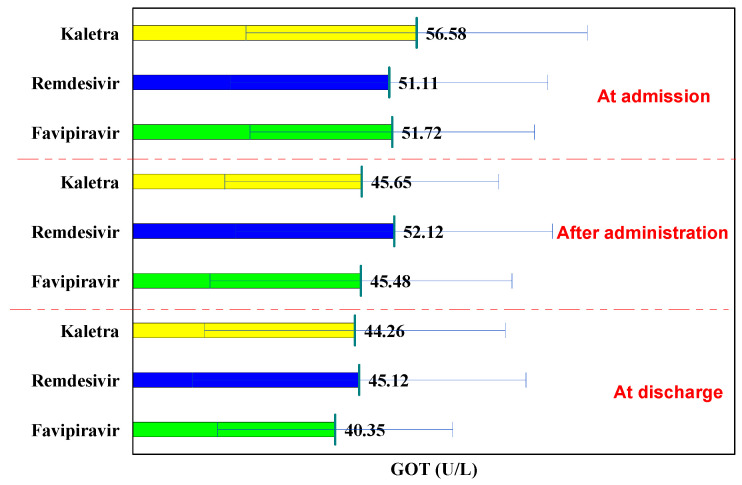
GOT evolution.

**Figure 3 biology-11-00013-f003:**
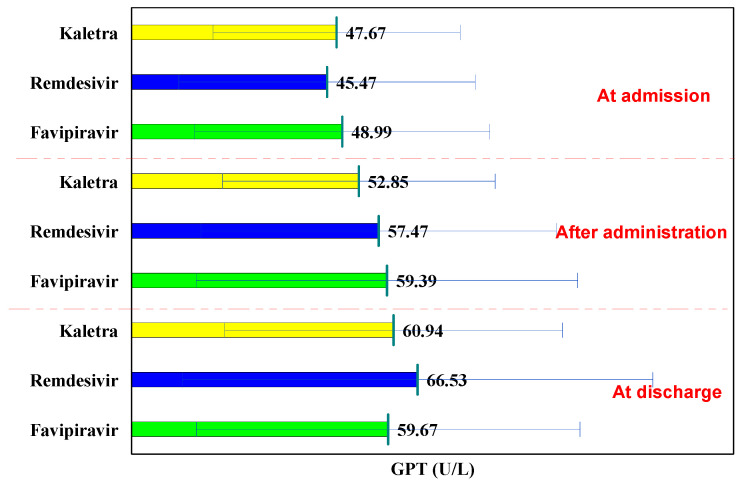
GPT evolution.

**Figure 4 biology-11-00013-f004:**
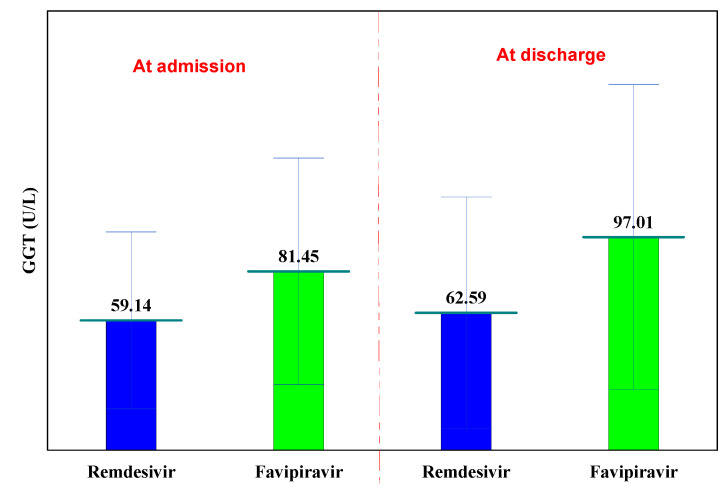
GGT evolution.

**Figure 5 biology-11-00013-f005:**
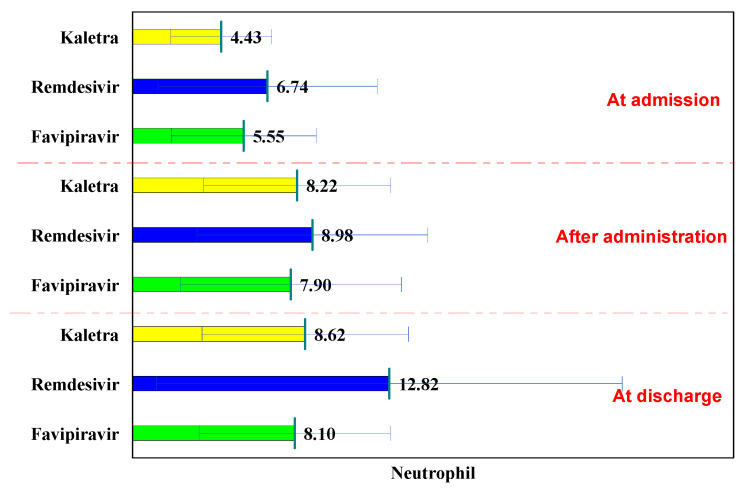
Neutrophils evolution.

**Figure 6 biology-11-00013-f006:**
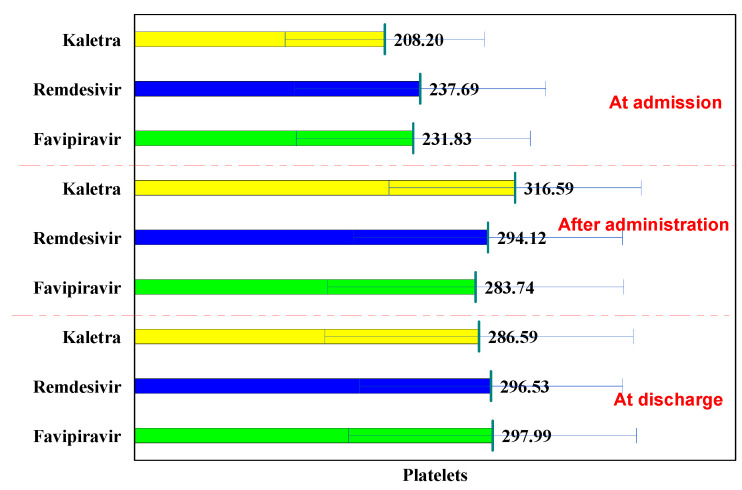
Platelet evolution.

**Figure 7 biology-11-00013-f007:**
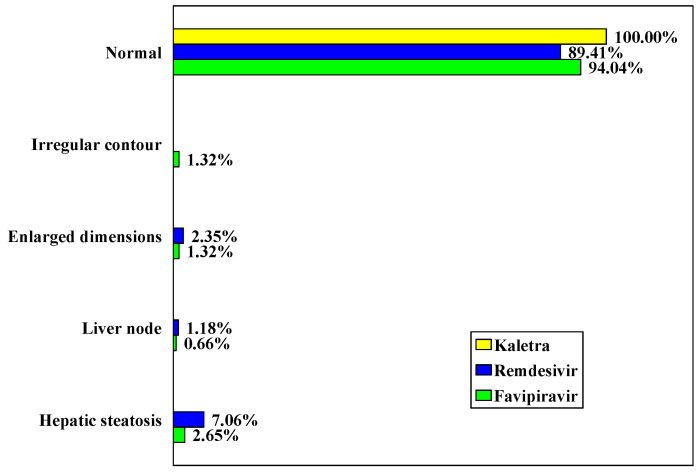
Distribution of cases according to CT image of the liver.

**Table 1 biology-11-00013-t001:** Characteristics of the population.

	No.	%	Min/Max	MD ± DS
Sex (F/M)	139/133	51.10/48.90		
Age (years)			26–88	60.18 ± 13.06
Environment (U/R)	152/120	55.88/44.12		

**Table 2 biology-11-00013-t002:** Distribution of cases according to the severity of the disease.

Severity of the Disease	Lopinavir/Ritonavir (Kaletra)		Remdesivir		Favipiravir	
Mild	3	8.33%	2	2.35%	18	11.92%
Moderate	18	50.00	13	15.29	50	33.11
Severe	15	41.67	70	82.35	83	54.97

**Table 3 biology-11-00013-t003:** GOT evolution.

GOT	at Hospitalization	after Administration	P^1−2^	at Discharge	P^1−3^	P^2−3^
Lopinavir/Ritonavir (Kaletra)	56.58 ± 34.01	45.65 ± 27.30	0.142	44.26 ± 29.99	0.112	0.843
Remdesivir	51.11 ± 31.65	52.12 ± 31.57	0.834	45.12 ± 33.27	0.231	0.161
Favipiravir	51.72 ± 28.36	45.48 ± 30.11	0.066	40.35 ± 23.41	<0.001	0.102
p^K-R^	0.412	0.268		0.892		
p^K-F^	0.432	0.975		0.480		
p^R-F^	0.882	0.117		0.246		

**Table 4 biology-11-00013-t004:** GPT evolution.

GPT	at Hospitalization	after Administration	P^1−2^	at Discharge	P^1−3^	P^2−3^
Lopinavir/Ritonavir (Kaletra)	47.67 ± 28.75	52.85 ± 31.72	0.477	60.94 ± 39.32	0.114	0.354
Remdesivir	45.47 ± 34.49	57.47 ± 41.36	0.042	66.53 ± 54.74	0.003	0.225
Favipiravir	48.99 ± 34.33	59.39 ± 44.40	0.025	59.67 ± 44.64	0.022	0.957
p^K-R^	0.719	0.514		0.535		
p^K-F^	0.812	0.322		0.869		
p^R-F^	0.453	0.740		0.328		

**Table 5 biology-11-00013-t005:** GGT evolution.

GGT	at Hospitalization	at Discharge	P^1−3^
Remdesivir	59.14 ± 40.27	62.59 ± 52.77	0.633
Favipiravir	81.45 ± 51.56	97.01 ± 69.41	0.437
p^R-F^	0.055	0.067	

**Table 6 biology-11-00013-t006:** Evolution of neutrophils.

Neutrophils	at Hospitalization	after Administration	P^1−2^	at Discharge	P^1−3^	P^2−3^
Lopinavir/Ritonavir (Kaletra)	4.43 ± 2.52	8.22 ± 4.67	<0.001	8.62 ± 5.15	<0.001	0.740
Remdesivir	6.74 ± 5.47	8.98 ± 5.75	0.010	12.82 ± 11.62	<0.001	0.007
Favipiravir	5.55 ± 3.62	7.90 ± 5.51	<0.001	8.10 ± 4.78	<0.001	0.737
p^K-R^	0.002	0.460		0.007		
p^K-F^	0.034	0.727		0.595		
p^R-F^	0.077	0.165		0.001		

**Table 7 biology-11-00013-t007:** Platelets evolution.

Platelets	at Hospitalization	after Administration	P^1−2^	at Discharge	P^1−3^	P^2−3^
Lopinavir/Ritonavir (Kaletra)	208.20 ± 82.90	316.59 ± 104.85	<0.001	286.59 ± 128.22	0.004	0.295
Remdesivir	237.69 ± 104.47	294.12 ± 111.79	0.001	296.53 ± 109.53	<0.001	0.887
Favipiravir	231.83 ± 97.34	283.74 ± 123.27	<0.001	297.99 ± 119.79	<0.001	0.316
p^K-R^	0.106	0.304		0.692		
p^K-F^	0.148	0.117		0.639		
p^R-F^	0.673	0.513		0.925		

**Table 8 biology-11-00013-t008:** Distribution of cases according to CT image of the liver.

	Lopinavir/Ritonavir (Kaletra)		Remdesivir		Favipiravir	
Normal	36	100.00	76	89.41	142	94.04
Irregular contour	0	0.00	0	0.00	2	1.32
Increased dimensions	0	0.00	2	2.35	2	1.32
Liver nodules	0	0.00	1	1.18	1	0.66
Hepatic steatosis	0	0.00	6	7.06	4	2.65

## Data Availability

Not applicable.

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
