# Peer review of "Evaluation of Hepatic Biochemical Parameters during Antiviral Treatment in COVID-19 Patients"

_biology, 2021, doi:10.3390/biology11010013_

Round 1

Reviewer 1 Report

The manuscript entitled “Evaluation of hepatic biochemical parameters during antiviral treatment in Covid-19 patients by Marc Felicia et al. analyzes the hepatic function and some other biochemical parameters in COVID-19 patients undergoing treatment with different antiviral medications.

Unfortunately, the manuscript does not provide any additional information than what is already known in the literature regarding these drugs when administered to COVID-19 patients.

The information provided in the different graphs should be condensed into 1 single table. It is not necessary to show every demographic and biochemical parameter in graphs. This info is usually provided in the table format. There is no clear conclusion apart from what is already known.

The Introduction should be more focused. The methods should be more descriptive, for instance, what definition did the authors use for the classification of disease severity in their population? This is not described anywhere in the manuscript. The discussion is extremely short and does not provide any discussion points about their findings.

The figures should not be presented for a manuscript in 3D format. The format the authors used is good for a presentation, but not for a manuscript.

Author Response

The authors acknowledge the useful observations and suggestions of the reviewer’s as concerns the manuscript entitled first “Evaluation of hepatic biochemical parameters during antiviral treatment  in Covid-19 patients byFelicia Marc 1, Corina Moldovan2, Anica Hoza3, Patricia Restea4, Liliana Sachelarie5,*, Laura Romila6*,Corina Suteu7, Dorina Maria Farcas8

According to the reviewer’s recommendations, the suggestions were carefully considered, as follows:

Unfortunately, the manuscript does not provide any additional information than what is already known in the literature regarding these drugs when administered to COVID-19 patients.

         In Covid 19 disease, a number of drugs have been used as repurposed drugs, and the relationship between these drugs and potential liver disturbancies caused by them was not clear

          In a recent article (32) it is stated that is important to evaluate the potential liver damage caused by various drugs in order to help guide clinical practice. In this review, these treatments were associated with minimal liver function abnormalities, but it is very important to pay attention to multimedication.

 In fact, we found referrences about Lopinavir/Ritonavir and Remdesivir, but little referrences about Favipiravir. Also, we made a comparison between the 3 antivirals and the effect of each one on liver parameters ( hepatocytolisis and cholestasis).

These findings may be useful in clinical practice, offering trust and guidance. Our experience is small but  onest.

  1. The information provided in the different graphs should be condensed into 1 single table. It is not necessary to show every demographic and biochemical parameter in graphs. This info is usually provided in the table format.

      We inserted a table with the characteristics of the population at results.

  1. The Introduction should be more focused.

 The antiviral treatment started to be largely used in 2020, although it is not specific for Sars-cov-2 virus. From a chronologic point of view, at the beginning a combination of 2 antiretrovirals was used (Lopinavir/Ritonavir), then Remdesivir and Favipiravir.

However, there were some concerns about potential liver damage or other side-effects.

Especially the combination of Lopinavir/ Ritonavir was strongly chriticised that is not effective in Covid 19 disease and also has side-effects, one of them being altering of hepatic function. In our department of internal medicine we had many patients who refused this therapy and signed in their medical file. Also there were some reserves for Favipiravir and Remdesivir , as they were newly introduced drugs , with not too much previous experience in coronavirus disease.

  1. The methods should be more descriptive, for instance, what definition did the authors use for the classification of disease severity in their population? This is not described anywhere in the manuscript.

 In order to establish the disease' severity,patients were evaluated from a clinical poin of view and also with biochemical analysis and thoracic, abdominal, cranial CT ( as needed).  From the clinical point of view, mild forms were with no oxigen requirements, while severe forms required high flow oxigen support.

 From the paraclinical point of view, to determine the severity of the disease, whether its mild, moderate or severe and also if it is viral or mixed Covid-19 pneumonia (viral and bacterial),the most important criteria was the degree of lung involvment described at computer tomography and the value of biochemical parameters characteristic for this coronavirus (C reactive protein,ESR =erythrocite sedimentation rate,lactate dehydrogenase, creatinkinase, limphocytes, plateletes,D-dimers). Also damage to other organs and systems caused by coronavirus, and the presence of co-morbidities were used in order to classify the severity of the disease.

  1. The discussion is extremely short and does not provide any discussion points about their findings.

                       Liver plays a major role in many diseases and it proves to be important also in CovidThere are several theories proposed for liver damage in Covid 19, such as direct effect of        the virus on hepatocytes or biliary epithelium via Angiotensin- converting enzyme (ACE2) receptors expression, liver injury related to increased immune response (Cytokine storm) and immune mediated damage, drug toxicity (Acetaminophen, antivirals or Hydroxychloroquine) and liver failure occuring in patients having multiorgan dysfunction.  (21,22,23,24). 

                          In our study we excluded patients with acute liver injury and elevated liver enzymes at admission, these findings being associated with increased severity and poor outcome.

 The drugs used in treating Covid 19 disease were initially intended for other diseases (viruses). Various drugs are being used as repurposed drugs, as there is no specific drug or effective treatment strategy against Covid 19. Multiple challenges associated with repurposed drugs have been identified, including dose adjustments, route of administration, acute/chronic toxicity .

   The relationship between the drugs used for Covid treatment and liver disturbance remains controversial. It is essential to evaluate the potential liver damage caused by various drugs in order to help guide clinical practice. (25,26,27,28,29).

 In our study,antiviral treatment was offered to an approximately equal number of women and men (139 women and 133 men), with a mean age of 59.95 years.

The low number of patients has several explanations; for example, there was a strong opinion in population that the combination Lopinavir/Ritonavir is not effective against the coronavirus Sars-cov 2 and some patients refused the administration of this drug and signed in their consent. Also there were some reserves against Favipiravir and Remdesivir, because they were newly introduced drugs; another serious problem was that patients arrived at the hospital after ten or more days of treatment at home and antiviral therapy was not useful anymore. Patients with high levels of aminotransferases were excluded from the study.

Chronologically, Lopinavir/Ritonavir (Kaletra) , Remdesivir, Favipiravir treatments were administrated to patients with Covid-19 pneumonia. The treatment was performed according to the recommendations of the Ministry of Health protocols in force at that time. The Lopinavir-Ritonavir treatment was withdrawn from the protocol, but the other two drugs remained as a recommendation. Remdesivir was used for the treatment of severe forms of the disease, and Favipiravir for mild and moderate forms.

Due to warnings that antiviral therapy may lead to impaired liver function  increased attention has been provided to hepatic parameters during the treatment including the 3 antiviral drugs, therefore, that parameters indicating hepatocytolysis and cholestasis have been monitored more frequently.

 In an randomized controlled study on Lopinavir/Ritonavir treated adult patients hospitalized with mild/moderate disease, only one patient developed elevation over 2,5 fold above the normal limit. (30)

In our study,GOT (aspartat aminotransferase) did not increase and GPT (alanin aminotransferase) increased a from a median value of 47,67 U/L to 60,94 U/L. Also, this combination doesn't influence the values of neutrophils and platelets.

                             The association of Remdesivir with liver injury remains uncertain, and is essential to evaluate the safety of this drug. One randomized multicenter trial with 237 patients (31) found similar aminotransferases levels between Remdesivir and placebo group. In our study,GOT did not increase, but the value of GPT increased from a median value of 45,4 U/L to 66,5 U/L. We also observed a slight elevation of cholestasis - gammaglutamil transpeptidase increased from 59,1 to 62,5 U/L and no negative effect on the number of neutrophils and plateletes.

In a recent article (32) it is stated that is important to evaluate the potential liver damage caused by various drugs in order to help guide clinical practice. In this review, these treatments were associated with minimal liver function abnormalities, but it is very important to pay attention to multimedication. In fact, we found referrences about Lopinavir/Ritonavir and Remdesivir, but little referrences about Favipiravir. Also, we made a comparison between the 3 antivirals and the effect of each one on liver parameters.

In our study, Favipiravir did not alter the value of GOT; GPT registered a change from 48,9 U/L to 59,6 U/L and also cholestasis enzyme increased - from 81,4 to 97,0 U/L. The levels of neutrophils and plateletes were not affected.

  1. There is no clear conclusion apart from what is already known.

Conclusions

From the viewpoint of the hepatic function damage (hepatocytolysis syndrome), the treatment with the 3 antiviral drugs did not lead to an increase in GOT values. Instead, the value of GPT increased after the use of each of the 3 drugs, but only slightly, no doubling of the values being recorded with any of the drugs. We observed that Remdesivir has the highest influence on the value of GPT, but not in a dangerous manner. In fact, no patient was withdrawn from the therapy.

 Regarding the cholestasis syndrome, the GGT value increased slightly, statistically insignificant, under the influence of treatments with Favipiravir and Remdesivir. Favipiravir caused a more pronounced cholestasis syndrome than Remdesivir.

The parameters related to the full blood count - the neutrophils and platelets, which could have been modified by the antiviral treatment, were not negatively influenced by the use of the 3 drugs, so no neutropenia or thrombocytopenia was observed.

 The use of the 3 antiviral drugs did not cause major liver damage and, clinically, in most cases, the Covid-19 viral pneumonias had a favorable evolution.

 These findings may be useful for medical community, offering trust, adding more information about the fact that these antiviral drugs, although not specific, are not so dangerous as they seemed to be at the beginning of the pandemics

  1. The figures should not be presented for a manuscript in 3D format. The format the authors used is good for a presentation, but not for a manuscript.

          The figures have been changed

Thank you very much for all your suggestions and good advice!

I remain most respectfully yours,

Prof.dr. Liliana Sachelarie

Reviewer 2 Report

bbreviations such as RT-PCR, GOT and GPT are used in the abstract, but their full names are not written. This error is also made in the text. These errors need to be fixed.
I couldn't see the result section in the summary section. Can you check this?
The "Background" part of the summary is too long. This section should be shortened to a maximum of two sentences.

In the material method section, it should be explained which statistical analysis program is used and which methods are used for analysis. Therefore, the "statistical analysis" section should be added to the article.

I recommend that they benefit from the following articles in the introduction of the article and I suggest that these articles be added to the reference list.

Sahin TT, Akbulut S, Yilmaz S. COVID-19 pandemic: Its impact on liver disease and liver transplantation. World J Gastroenterol. 2020 Jun 14;26(22):2987-2999.

Altunisik Toplu S, Bayindir Y, Yilmaz S, Yalçınsoy M, Otlu B, Kose A, Sahin TT, Akbulut S, Isik B, BaÅŸkiran A, Koc C. Short-term experiences of a liver transplant centre before and after the COVID-19 pandemic. Int J Clin Pract. 2021 Oct;75(10):e14668. 

Author Response

The authors acknowledge the useful observations and suggestions of the reviewer’s as concerns the manuscript entitled first “Evaluation of hepatic biochemical parameters during antiviral treatment  in Covid-19 patients byFelicia Marc 1, Corina Moldovan2, Anica Hoza3, Patricia Restea4, Liliana Sachelarie5,*, Laura Romila6*,Corina Suteu7, Dorina Maria Farcas8

According to the reviewer’s recommendations, the suggestions were carefully considered, as follows:

Abreviations such as RT-PCR, GOT and GPT are used in the abstract, but their full names are not written. This error is also made in the text. These errors need to be fixed

    Abbreviations: RT-PCR (reverse transcription polymerase chain reaction), GOT (Aspartat aminotransferase), GPT (alanin aminotransferase) GGT (gammaglutamil-transpeptidase), ALP (alkaline phosphatase).

I couldn't see the result section in the summary section. Can you check this?

(3). Results: In the group of studied patients, the mean value of Aspartat aminotransferase did not increase above normal at discharge, Alanin aminotransferase increased, but below twice the normal values, and cholestasis registered a statistically insignificant slight increase. (4) Conclusions: In our study, we found that all the 3 antivirals were generally well tolerated and their use did not alter liver function in a significant manner.

The "Background" part of the summary is too long. This section should be shortened to a maximum of two sentences.

(1) Background: The antiviral treatment for Covid19 disease started to be largely used in 2020 and is efficient, although it is not specific for Sars-Cov-2 virus. There were some concerns that it may produce liver damage or other side effects.

In the material method section, it should be explained which statistical analysis program is used and which methods are used for analysis. Therefore, the "statistical analysis" section should be added to the article.

Data processing was performed using the SPSS 20 program. Frequency ranges, average parameter values, standard deviations were calculated. Tests of statistical significance by the c2 method were used, and ANOVA (Brown-Forsythe) was used to compare the means. The level of statistical significance was 0.05.

I recommend that they benefit from the following articles in the introduction of the article and I suggest that these articles be added to the reference list.
1.Sahin TT, Akbulut S, Yilmaz S. COVID-19 pandemic: Its impact on liver disease and liver transplantation. World J Gastroenterol. 2020 Jun 14;26(22):2987-2999.

2.Altunisik Toplu S, Bayindir Y, Yilmaz S, Yalçınsoy M, Otlu B, Kose A, Sahin TT, Akbulut S, Isik B, BaÅŸkiran A, Koc C. Short-term experiences of a liver transplant centre before and after the COVID-19 pandemic. Int J Clin Pract. 2021 Oct;75(10):e14668. 

 Bibliography  title nr.23 and 24

Thank you very much for all your suggestions and good advice!

I remain most respectfully yours,

Prof.dr. Liliana Sachelarie

Reviewer 3 Report

Proposed paper is interesting and well written-. However, some revisions are needed before it can be accepted for pubblication:

  • A general table with all the characetristic of the whole population (not divided by antiviral therapies) should be provided in order to correctly understand the population.
  • The low number of patients should be discussed in the relative section.
  • The mortality analysis should be deleted. In fact it could seems that some antiviral determined a greater protection than others but this could not be determined by a paper with few patients like this. Please remove this analysis.
  • Epatic involvment during COVID-19 infection could be related to severe sepsis more than to antiviral specific drugs. All the significant founding need to be inserted into a multivariate model with disease severity, GFR, age and so on as covariates in order to confirm it.
  • When speaking on CV risk of COVID-19 please cite also this recently published paper: High Blood Press Cardiovasc Prev. 2021 Sep;28(5):439-445.

Author Response

The authors acknowledge the useful observations and suggestions of the reviewer’s as concerns the manuscript entitled first “Evaluation of hepatic biochemical parameters during antiviral treatment  in Covid-19 patients byFelicia Marc 1, Corina Moldovan2, Anica Hoza3, Patricia Restea4, Liliana Sachelarie5,*, Laura Romila6*,Corina Suteu7, Dorina Maria Farcas8

According to the reviewer’s recommendations, the suggestions were carefully considered, as follows:

  • A general table with all the characteristics of the whole population (not divided by antiviral therapies) should be provided in order to correctly understand the population.

       A table with the characteristic of the whole population, not divided by antiviral therapy is at the beginning of results.

  • The low number of patients should be discussed in the relative section.

The low number of patients has several explanations; for example, there was a strong opinion in population that the combination Lopinavir/Ritonavir is not effective against the coronavirus Sars-cov 2 and some patients refused the administration of this drug and signed in their consent. Also there were some reserves against Favipiravir and Remdesivir, because they were newly introduced drugs; another serious problem was that patients arrived at the hospital after ten or more days of treatment at home and antiviral therapy was not useful anymore. Patients with very elevated level of aminotransferases were excluded from the study.

  • The mortality analysis should be deleted. In fact it could seems that some antiviral determined a greater protection than others but this could not be determined by a paper with few patients like this. Please remove this analysis.

      Mortality analysis was deleted

  • Epatic involvment during COVID-19 infection could be related to severe sepsis more than to antiviral specific drugs. All the significant founding need to be inserted into a multivariate model with disease severity, GFR, age and so on as covariates in order to confirm it.

   We did not have patients with severe sepsis; they were usually transferred to the Intensive care unit as their clinical status worsened and procalcitonin levels were very high

  • When speaking on CV risk of COVID-19 please cite also this recently published paper: High Blood Press Cardiovasc Prev. 2021 Sep;28(5):439-445.

 Bibliography title nr.8

Thank you very much for all your suggestions and good advice!

I remain most respectfully yours,

Prof.dr. Liliana Sachelarie

Round 2

Reviewer 1 Report

The revised manuscript entitled “Evaluation of hepatic biochemical parameters during antiviral treatment in Covid-19 patients by Marc Felicia et al. analyzes the hepatic function and some other biochemical parameters in COVID-19 patients undergoing treatment with different antiviral medications.

While the authors have tried to improve the manuscript, unfortunately, it lacks novelty, and the data is not accurately presented.

Among the critical issues that this manuscript has is the inclusion and exclusion criteria definition. It is not clear from the description exactly what criteria were used.

There is no information regarding the demographics of the patients.

The definition used by the authors for the severity is extremely vague. “From the clinical point of view, mild forms were with no oxigen requirements, while severe forms required high flow oxigen support”.  This is not an accepted definition for severity.

Again, I emphasize the need to condense all the results into a single table.

There is a need to perform a major revision on the statistics. There is no description of how the data were evaluated for normality. There is no description of how the data is being presented and depending on the variables how is being compared?

The discussion is not focused yet on discussing their results.

The manuscript needs an extensive English revision. It contains several grammatical errors and typos throughout.

Reviewer 2 Report

Thank you for this nice paper. 

Reviewer 3 Report

Authors replies to all the query raised and paper improves and can now be accepted for pubblication.